# Investigation of Thermal Stability and Reactivity of Rh Nanoclusters on an Ultrathin Alumina Film

**Zhen-He Liao [1], Po-Wei Hsu [1], Ting-Chieh Hung [1], Guan-Jr Liao [1], Zhao-Ying Chern [2], Yu-Ling Lai [3], Li-Chung Yu [3], Yao-Jane Hsu [3], Jeng-Han Wang [2], Peilong Chen [1] and Meng-Fan Luo [1],***

[1] Department of Physics, National Central University, 300 Jhongda Road, Jhongli District, Taoyuan 32001, Taiwan; jason23147@gmail.com (Z.-H.L.); p761021@hotmail.com (P.-W.H.); gunhoogunsd1@gmail.com (T.-C.H.); liaojohnnylkz01@gmail.com (G.-J.L.); peilong@phy.ncu.edu.tw (P.C.)

[2] Department of Chemistry, National Taiwan Normal University, No. 88, Sec. 4, Ting-Chow Rd, Taipei 11677, Taiwan; zying127@outlook.com (Z.-Y.C.); jenghan@ntnu.edu.tw (J.-H.W.)

[3] National Synchrotron Radiation Research Center, 101 Hsin-Ann Road, Hsinchu Science Park, Hsinchu 30076, Taiwan; yllai@nsrrc.org.tw (Y.-L.L.); yu.lc@nsrrc.org.tw (L.-C.Y.); yjhsu@nsrrc.org.tw (Y.-J.H.)

\* Correspondence: mfl28@phy.ncu.edu.tw

**Abstract:** We studied the structural and morphological evolution of Rh clusters on an ordered ultrathin alumina film grown on NiAl(100) in annealing processes, under ultrahigh vacuum conditions and with various surface probe techniques. The Rh clusters, prepared on vapor deposition of Rh onto the alumina film at 300 K, had an fcc phase and grew in the (100) orientation; the annealing altered the cluster structure little—the lattice parameter decreased by a factor <2%—but the cluster morphology significantly. With elevated temperature, small clusters (diameter ≤1.5 nm) decreased little in size; in contrast, large clusters (diameter ≥2.0 nm) varied in a complex manner—their mean diameter decreased to about 1.5 nm on annealing to 450 K, despite their similar height, while it increased to above 2.0 nm at temperature ≥570 K. This atypical decrease in size was governed predominantly by energetics. Such a reduced size enhanced the total surface area as well as the reactivity of the clusters toward methanol decomposition, so increased the production of $D_2$ ($H_2$) and CO from decomposed methanol-$d_4$ (or methanol). The result implies a higher temperature tolerance for Rh clusters on the alumina film and a practical approach to prepare small Rh clusters with high reactivity.

**Keywords:** Rh nanoclusters; $Al_2O_3$; size effect; methanol decomposition

## 1. Introduction

As the reactivity of catalysts is largely associated with their structures [1–5] and as catalyzed reactions could proceed at elevated temperature, a knowledge of thermal stability of such structures and how they evolve with elevated temperature becomes desirable. Preceding work showed that elevated temperature can alter the morphologies, sizes and structures of oxide-supported metal nanoclusters, typical catalysts [1–14]; it can also induce mass transport and thus encapsulation and oxidation of the oxide-supported nanoclusters [1–5,11,15–20]. The objective of the present work is to study the effect of elevated temperature on the structure and morphology of oxide-supported rhodium (Rh) clusters and examine the effect with catalyzed decomposition of methanol (methanol-$d_4$), which is the principal reaction applied in direct methanol fuel cells (DMFC) [21–25] and also serving as a source of hydrogen. Rh-based catalysts have been extensively used; Rh is alloyed with platinum (Pt, the primary catalyst for methanol reactions) catalysts to improve catalytic properties; adsorption and decomposition of methanol on Rh single crystal surfaces [26–29] have hence been extensively studied but the methanol

reactions on oxide-supported Rh clusters [30,31], a realistic model system and how they are affected by thermally-induced structural changes are little investigated. The present work aims to remedy this lack and to acquire insight into the effect.

We investigated the structural and morphological evolution of Rh clusters supported on an ordered ultrathin alumina film with temperature and its effect on the decomposition of methanol ($CH_3OH$) and methanol-$d_4$ ($CD_3OD$). The alumina film was grown on oxidation of NiAl(100) (denoted as $Al_2O_3$/NiAl(100)) and the experiments were performed under ultrahigh vacuum (UHV) conditions and with various techniques to probe the surface. The Rh clusters were prepared by depositing a vapor onto $Al_2O_3$/NiAl(100) at 300 K and annealing to selected temperatures. We characterized the morphology and structure of the Rh clusters with scanning tunneling microscopy (STM) and reflection high-energy electron diffraction (RHEED) and monitored the catalyzed reactions with temperature-programmed desorption (TPD) and synchrotron-based photoelectron spectroscopy (PES). The morphological results were complemented with Monte Carlo simulation, for which the relative rates of kinetic processes were justified with their activation energies derived from density-functional-theory (DFT) calculations, to explore the thermal effect on the structuring of the Rh clusters.

The Rh clusters, as prepared, were at nanoscale; they were structurally ordered, exhibiting an fcc phase and growing in the (100) orientation (denoted as Rh(100) clusters). The annealing altered little their structures but dramatically their morphologies. We found that the diameter of large clusters (diameter $\geq 2.0$ nm) decreased to a preferential one about 1.5 nm on annealing to 450 K, despite the height remained similar. Although the morphological evolution is governed by both kinetics and energetics, this atypically decreased size is predominated by the latter [10,32]. The reduced size increased the total surface area and also promoted the reactivity of the clusters toward methanol decomposition [31]. The size-dependent reactivity has been found for varied transition metal catalysts and reactions, such as Au clusters toward various reactions (oxidation of CO, oxidation and hydrochlorination of hydrocarbons, water-gas shift and NO reduction) [33–36], Rh toward CO dissociation [37,38], Ni toward ethane hydrogenolysis [39,40] and Cu toward $N_2$ formation from NO [5,41]. The control over the cluster size thus becomes essential in manipulating catalytic properties of catalysts. The present result implies a facile approach to prepare small Rh clusters with high reactivity and also an advantage for the Rh clusters on $Al_2O_3$/NiAl(100) that they can tolerate an elevated reaction temperature to a certain extent, with no decreased reactivity.

## 2. Results and Discussion

### 2.1. Morphology and Density of the Supported Rh Clusters

The morphology and density of the Rh clusters, grown from the deposition of Rh vapor onto $Al_2O_3$/NiAl(100) at 300 K, were characterized with STM. Their response to elevated temperature was shown to depend on the cluster size. Figure 1a–d exemplify the evolution of the size and density of smaller Rh clusters/$Al_2O_3$/NiAl(100) with annealing temperature; the insets in the figures show, for each temperature, the characteristic histograms of height and diameter of the clusters; a Gaussian fit to the size distribution is also shown in each histogram. At 300 K, the 0.5 monolayer equivalent (MLe) Rh clusters had a mean diameter about 1.4 nm and height about 0.6 nm (Figure 1a); the clusters were largely aligned as the protrusions stripes, arising from a lattice mismatch between the $Al_2O_3$ film and NiAl(100) [42–44], are preferential nucleation sites, as observed earlier for other deposited transition metals [44–46]. With the temperature increased to 450 K, both the mean diameter and height altered little (Figure 1b); the cluster density also remained similar ($1.66 \times 10^{13}$ cm$^{-2}$). On increasing the temperature further to 570 K, the mean diameter decreased slightly to 1.2 nm whereas the height increased to 0.67 nm (Figure 1c); the cluster density decreased to $1.44 \times 10^{13}$ cm$^{-2}$ and the total amount of Rh also decreased to about 0.4 MLe. The increased temperature facilitated the interlayer transportation in the clusters so the clusters became more three-dimensional; concomitantly, the diffusion of Rh into the substrate also started. Continuing the annealing to 800 K, both the diameter

and height decreased and the cluster density decreased notably to $1.03 \times 10^{13}$ cm$^{-2}$—only 0.17 MLe remained on the surface. At such a high temperature, the Rh atoms were readily dissociated from the clusters and diffused rapidly on the surface. The Ostwald ripening or conventional sintering did not occur because a great proportion of Rh atoms (dissociated from the clusters) diffused into the substrate, rather than joining an existing clusters or nucleating into a new cluster.

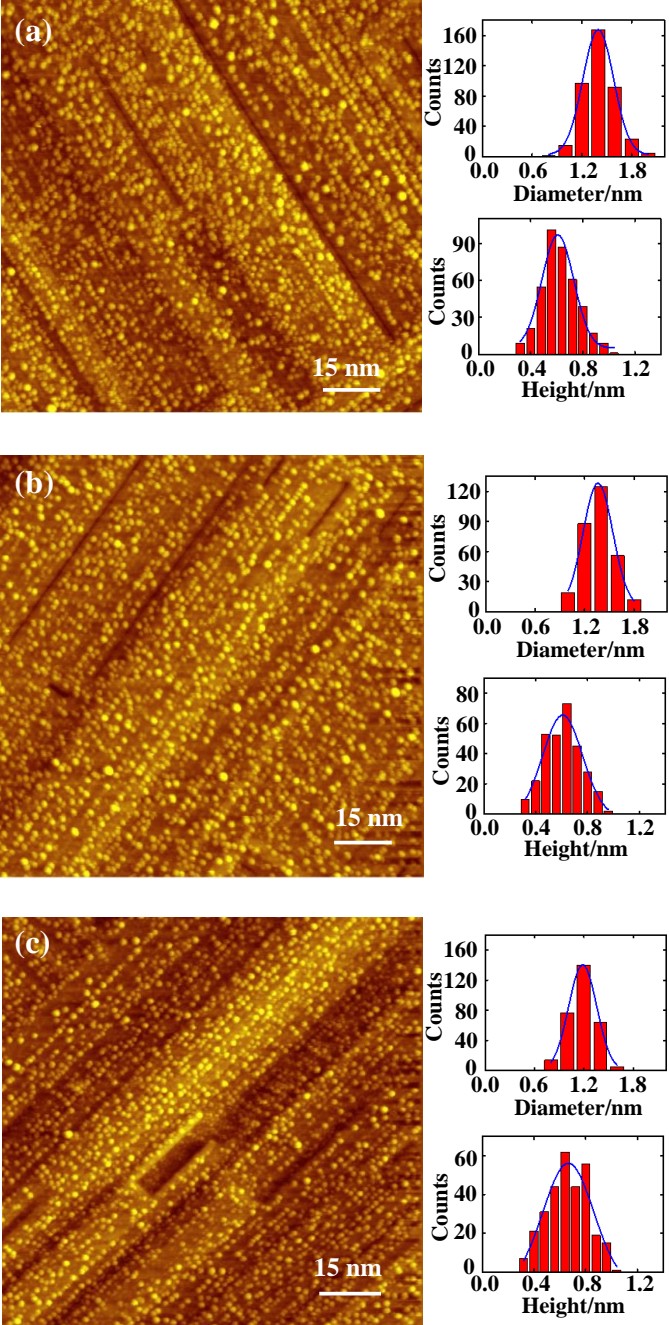

**Figure 1.** *Cont.*

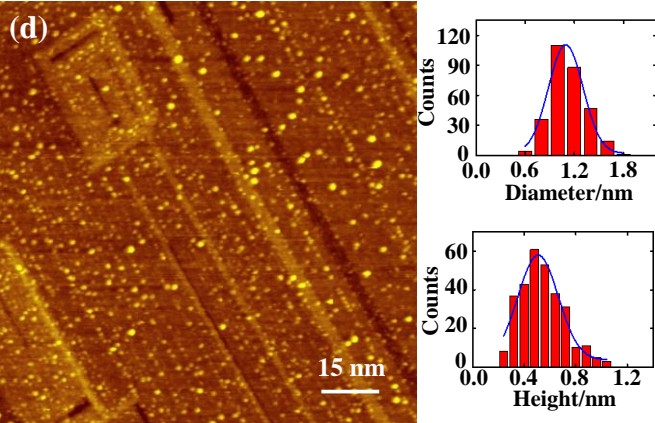

**Figure 1.** STM images for 0.5-MLe Rh deposited on a thin film of Al$_2$O$_3$/NiAl(100) at (**a**) 300 K and annealed stepwise to (**b**) 450, (**c**) 570 and (**d**) 800 K. The insets in (**a**–**d**) show characteristic histograms of height and diameter for each temperature; the curves are the best Gaussian fits to the distributions.

The larger clusters show a disparate response to the increased temperature. Figure 2a–d exemplify their morphological and density evolution with the annealing temperature. The 1.35-MLe Rh clusters, as prepared, had a mean diameter about 2.1 nm and height about 0.77 nm (Figure 2a), evidently larger than those at 0.5 MLe, but their density resembled that at 0.5 MLe ($1.65 \times 10^{13}$ cm$^{-2}$). The clusters' density was saturated at 0.5 MLe, so further deposited Rh did not nucleate into new clusters but joined existing ones and hence increased the cluster size significantly. On increasing the temperature to 450 K, the mean height remained similar (0.79 nm) whereas the diameter decreased evidently to 1.65 nm (Figure 2b); correspondingly, the cluster density increased to $2.26 \times 10^{13}$ cm$^{-2}$. As the total amount of Rh remained nearly the same, the diffusion of Rh into the substrate played no role in this morphological change. Increasing the temperature further, the clusters enlarged, the cluster density decreased and the total amount of Rh also decreased (Figure 2c,d). At 800 K, a typical Ostwald ripening was observed; both the mean diameter (2.4 nm) and height (0.88 nm) became greater than those of as-prepared clusters and the size distribution became broader—the numbers of both larger and smaller clusters were enhanced (Figure 2d). The diffusion of Rh into the substrate also became active; the total quantity of Rh decreased to about 0.64 MLe.

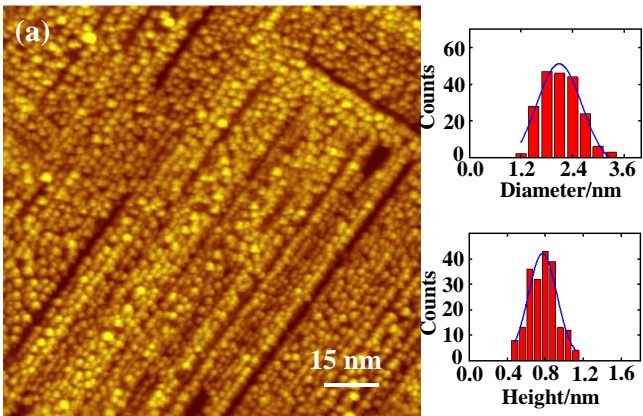

**Figure 2.** *Cont.*

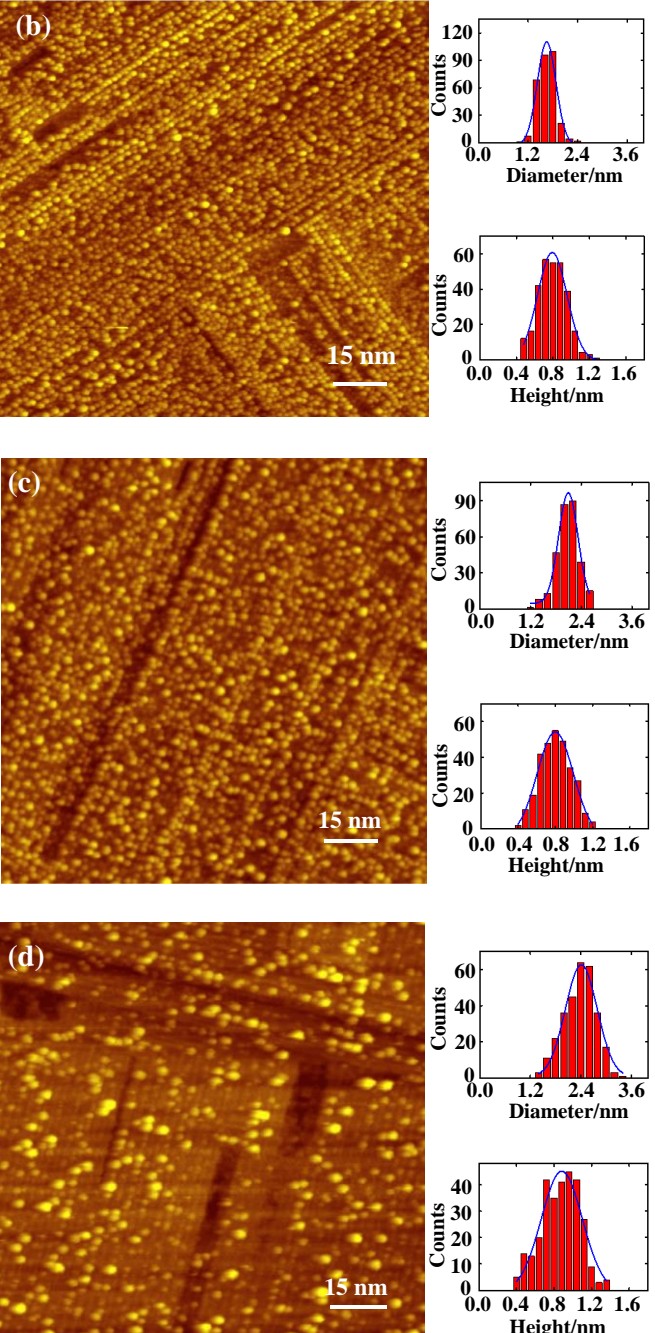

**Figure 2.** STM images for 1.35-MLe Rh deposited on a thin film of $Al_2O_3$/NiAl(100) at (**a**) 300 K and annealed stepwise to (**b**) 450, (**c**) 570 and (**d**) 800 K. The insets in (**a–d**) show characteristic histograms of height and diameter for each temperature; the curves are the best Gaussian fits to the distributions.

Figure 3a,b summarize the annealing-driven evolution of the diameter and height of Rh clusters at varied coverages; the error bars in the figures indicate the full width at half maximum of the best Gaussian fits to the distributions of diameter and height of the clusters for each temperature. Two trends are obviously shown for varied Rh coverages (sizes). For 0.5 MLe, the mean diameter continued to decrease from 1.4 nm to 1.1 nm and the mean height varied little below or at 570 K (0.6–0.67 nm) and decreased slightly to 0.5 nm at even higher temperatures (700 or 800 K). For 1.35- and 1.6-MLe Rh, the mean height altered only moderately with the temperature (mean height 0.77~0.88 nm), but the diameter decreased dramatically first at 450 K (from 2.1–2.2 nm to 1.5–1.6 nm) and then continued to increase with further elevated temperatures (2.2–2.4 nm). Figure 3c,d plot the corresponding variation

of the Rh quantity and cluster density with temperature. The quantity decreased evidently with temperature above 570 K (Figure 3c); at 800 K, the quantity decreased to below 50% of the original. The decrease reflects that Rh atoms from dissociated clusters diffused into the substrate, likely through surface defects [4,42–44]. Like the cluster diameter, the cluster density exhibited atypical behaviors. The cluster density for 0.5-MLe Rh decreased at all temperatures above 450 K whereas those for 1.35- and 1.6-MLe Rh increased unusually at 450 K and decreased at further elevated temperatures (Figure 3d). The increased cluster densities of 1.35- and 1.6-MLe Rh at 450 K imply that clusters became dissociated and new clusters formed, consistent with the concomitantly decreased cluster diameter (Figure 3a). Above 450 K, the clusters ripened but more Rh diffused into the substrate, so the diameter increased (Figure 3a) and the cluster density decreased (Figure 3d).

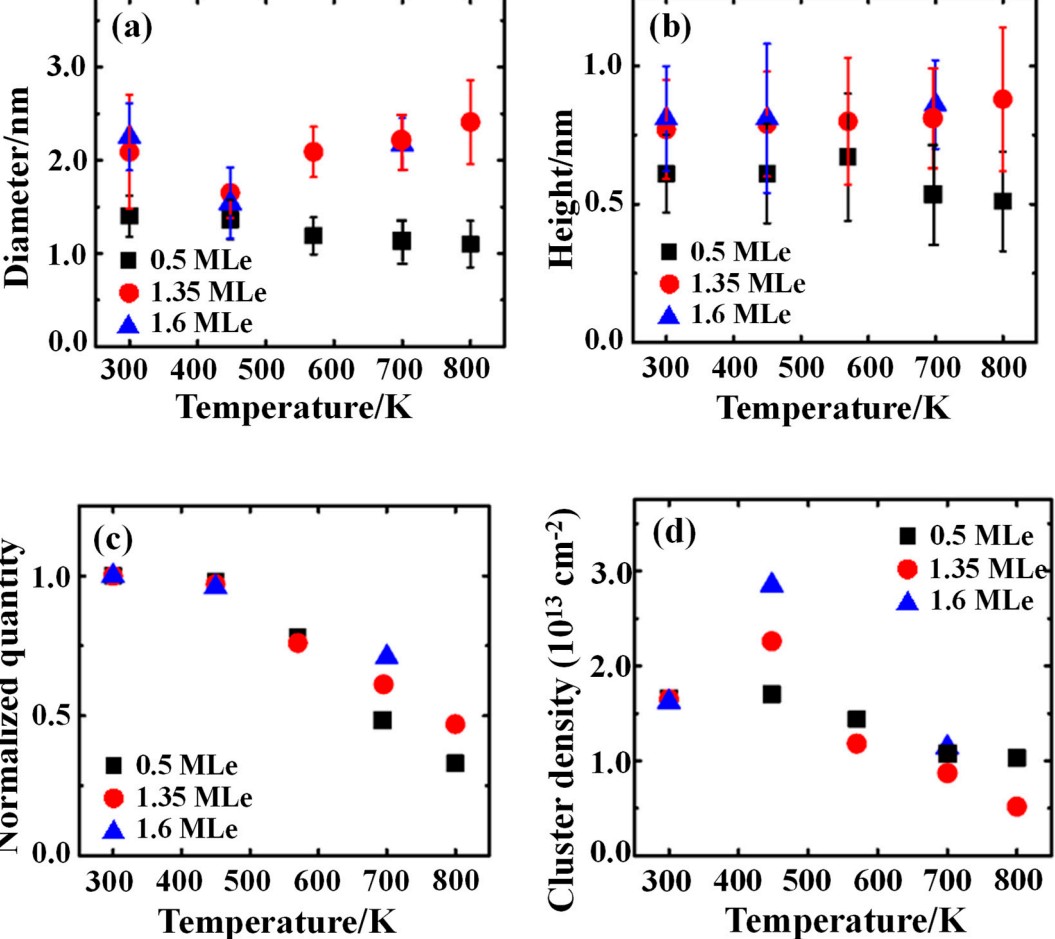

**Figure 3.** (**a**–**d**) Plot in order the evolution of the mean diameter, mean height, quantity and density of Rh clusters with temperature. Three Rh coverages, 0.5 (back square), 1.35 (red circle) and 1.6 MLe (blue triangle), are demonstrated. The errors bars in (**a**,**b**) indicate the full width at half maximum of the Gaussian curves which fit best the histograms of diameter and height of Rh clusters. The quantities in (**c**) were calculated with cluster densities and mean volumes of individual clusters for each Rh coverage; they were normalized to the quantities derived for clusters as prepared.

The atypical alteration of cluster size and density could result solely from a kinetic effect or involve an energetic one. The former argues an Ostwald ripening constrained by the channel of Rh diffusion to the substrate and facile formation of new clusters due to abundant preferential nucleation sites, the linear protrusions, on the $Al_2O_3$/NiAl(100) [10,44–46]. For 0.5-MLe Rh, the Rh atoms from dissociated Rh clusters diffused largely into the substrate above 570 K (instead of joining existing clusters) because of its small cluster density, so the cluster size decreased and no typical sintering

occurred. For 1.35- and 1.6-MLe Rh, as the linear protrusions limited the diffusion of Rh atoms (small diffusion length) and assisted the formation of new clusters (a small critical size for nucleation), due to a stronger interaction between the oxide defects and Rh [42–44,47,48], the Rh atoms detaching from Rh clusters nucleated readily into new clusters at 450 K, leading to the increased cluster density. At further elevated temperatures, the diffusion limitation was relieved and the critical size for nucleation increased—nucleation into new clusters became difficult, so more Rh atoms joined existing clusters, the clusters grew and cluster density decreased. The second possibility involves, in addition to the kinetic effect, an energetically preferential cluster size. We noted that the cluster diameter of 1.35- and 1.6-MLe Rh was decreased to a value (~1.5 nm) near that of 0.5-MLe Rh (Figure 3a) on annealing to 450 K; moreover, the cluster size at 0.5 MLe altered only a little with temperature, despite the cluster density and total amount evidently decreasing. The clusters with such a preferential diameter might be energetically more stable, as observed for Co clusters [10,32], so were not dissociated at 450 K. Our Monte Carlo modelling below verifies the involvement of the energetic effect.

## 2.2. Atomic Structures of the Supported Rh Clusters

The structural ordering of the Rh clusters is reflected in the RHEED patterns. Figure 4a–d exemplify the effect of elevated temperature on the structures of the clusters. The reflection rods of the RHEED patterns in the figures are ascribed to the oxide and to the NiAl(100) substrate; the half-order reflections in Figure 4a,c indicate the (2 × 1) structure of the ordered θ-Al$_2$O$_3$(100) [42,43,49,50]. The diffraction spots additionally superimposed on the reflection rods at azimuths [0–10] and [0–11] were attributed to structurally ordered Rh clusters. The patterns were observed at and above 1.0 MLe (Figure 4a,b), and became sharper when the temperature increased (Figure 4c,d). The variation implies that with annealing the content of the ordered structures was increased. Above 570 K, the patterns remained sharp but their intensities attenuated a little (Figure 4e–h), as the number of clusters decreased. The structurally ordered clusters grew in an fcc phase and had their (100) facets parallel to the surface of θ-Al$_2$O$_3$(100); besides, their [110] axes were parallel to direction [010] of the alumina surface, so Rh(100)[110]//Al$_2$O$_3$(100)[010]. The corresponding reciprocal lattice points at the two azimuths are drawn in Figure 4a,b. This orientation is favorable because the cluster's (100) facet matches better, structurally, the square oxygen lattice of the θ-Al$_2$O$_3$(100) surface [11,51,52]. The lattice parameter of the Rh clusters also increased to match the oxide surface. The mean lattice parameter of 1.0-MLe Rh clusters (mean diameter 1.8 nm and height 0.7 nm) was about 4.04 Å, increased by nearly 6% relative to bulk Rh (3.80 Å). The expansion decreased when the size or coverage was increased (Figure 4i), because the substrate effect diminished on the top few layers of the growing clusters. At 4.0 and 6.0 MLe, the Rh clusters already covered the whole alumina surface and their lower parts became merged. The annealing did not alter the cluster orientation but affected the lattice parameter a little. The annealed clusters have a trend of lattice parameters decreasing with increased temperature, in a scale ≤ 2% (Figure 4i). Such a decreased lattice parameter was typically ascribed to an attenuating substrate effect resulting from increased cluster sizes; nevertheless, the cluster size in the present case did not continue to increase (Figure 3). The result implies that when the clusters became more structurally ordered—the content of the ordered structures increased—with annealing, their average Rh-Rh distance decreased, toward their bulk value. Earlier studies observed negative thermal expansion for Pt nanocluster/γ-Al$_2$O$_3$ [53,54], while this effect is expected to be negligible in the present observation. Even if this negative thermal expansion could persist after the sample was cooled down to 300 K, the reported scale of the altered atom-atom distance is smaller (<1%) than the present one (Figure 4i).

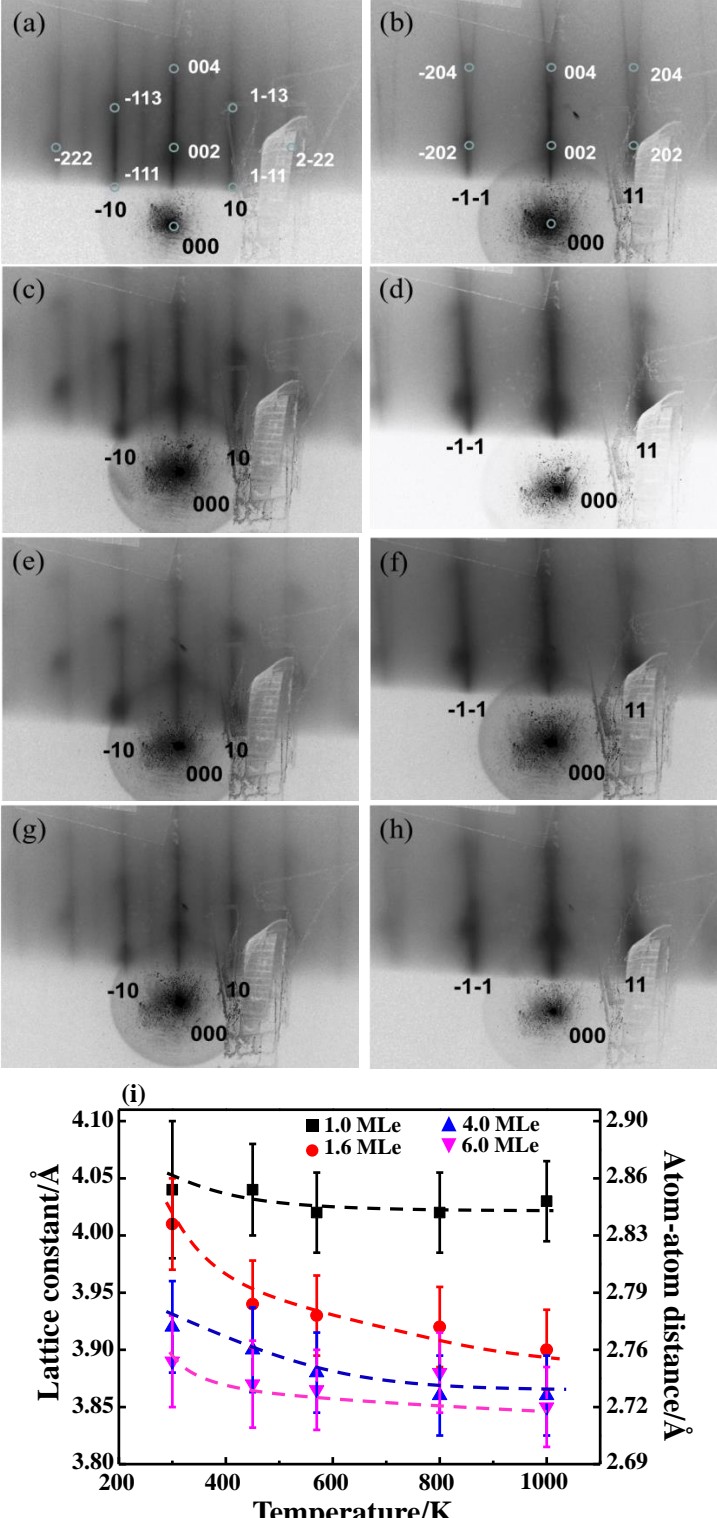

**Figure 4.** RHEED patterns for 1.6-MLe Rh clusters on Al$_2$O$_3$/NiAl(100) at (**a**,**b**) 300 K and annealed stepwise to (**c**,**d**) 450 K, (**e**,**f**) 570 K and (**g**,**h**) 800 K. (**a**,**c**,**e**,**g**) show patterns obtained at azimuth [0–10] of NiAl substrate, and (**b**,**d**,**f**,**h**) at azimuth [0–11]. Blue circles in (**a**,**b**) denote reciprocal-lattice points for the clusters of Rh(001)[110]//Al$_2$O$_3$(100)[010]. (**i**) plot of the lattice parameter of Rh clusters of varied sizes (coverages 1.0, 1.6, 4.0 and 6.0 MLe) as a function of temperature. The lattice parameters were derived by fitting the reciprocal-lattice nets to the diffraction spots. The errors bars were obtained based on the reproducibility and the size of the diffraction spots (the brightest areas, with top 2% in intensity).

### 2.3. Modelling of Annealing-Induced Morphological Alteration of Supported Rh Clusters

We conducted Monte Carlo simulation to illuminate whether or not the morphological change of the Rh clusters was solely attributed to thermally facilitated kinetic processes. An example of the modeled annealing-induced alteration is shown in Figure 5a. The lattice has $100 \times 100$ sites for each layer and the bonding energy $\epsilon$ is set as $e^{-\epsilon/kT} = 0.01$ for 300 K; the step count between each added atom is $M = 10$, the substrate defect binding parameter $h = 3$ (binding energy $3\,\epsilon$) and the coverage is 1.0 MLe. The figure exhibits the stable configurations of clusters, seen from top view, grown at 300 K and annealed to 570 and 800 K. We observed only minor changes when the temperature was raised, through 450 K, to 570 K; when increasing the temperature to 800 K, the cluster density decreased and size increased significantly—a ripening process. This trend (average diameter and height) is plotted against temperature in Figure 5b. The cluster size altered little below 700 K but increased remarkably at 800 K; the average diameter decreased slightly between 300 and 570 K as more atoms migrated from the bottom to upper layers, resulting in a concomitant increase of cluster height. Consistently, the cluster density also remained similar (60–70 in the lattice) below 700 K and decreased dramatically (to 30) at 800 K. The ripening proceeded evidently at 800 K under these kinetic conditions.

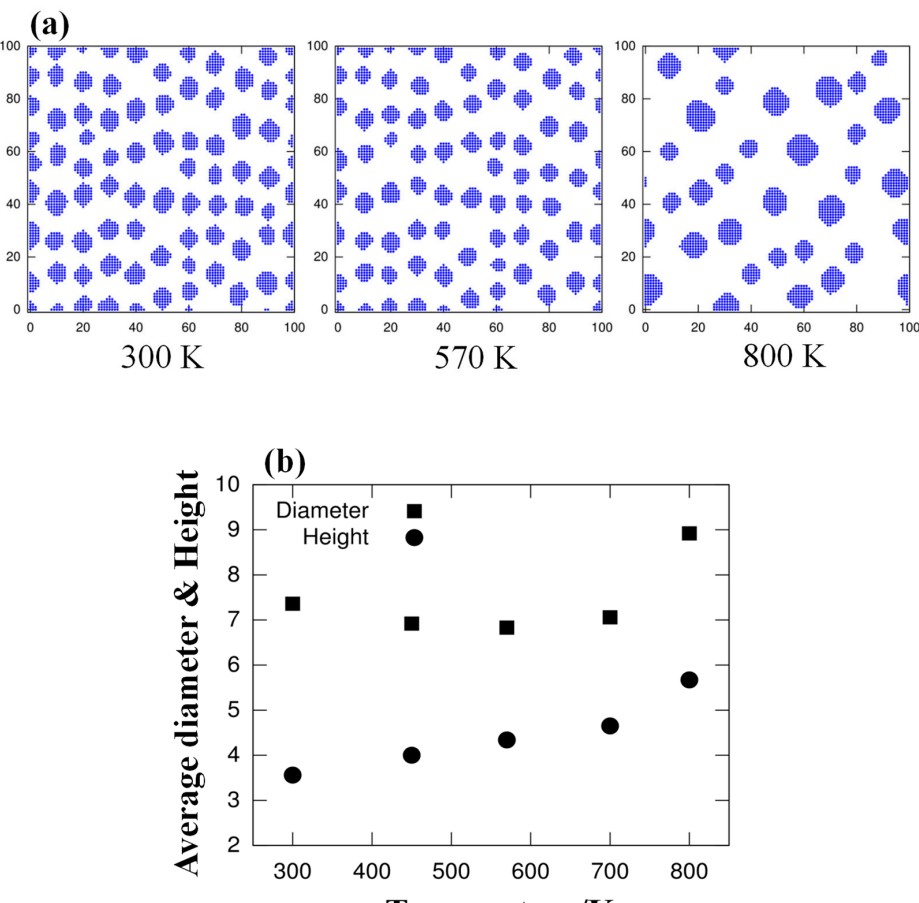

**Figure 5.** (**a**) Top view of the simulated configurations of Rh islands grown at 300 K and annealed to 570 and 800 K, as indicated. The lattice has $100 \times 100$ sites for each layer and the bonding energy $\epsilon$ is set as $e^{-\epsilon/kT} = 0.01$ for 300 K; the step count between each added atom is $M = 10$, the substrate defect binding parameter $h = 3$ and the coverage is 1.0 MLe. (**b**) plot of average diameter and height of simulated Rh islands as a function of temperature. The Rh islands were grown at 300 K and annealed to the temperatures as indicated.

Varied kinetic parameters, including the hopping rate, deposition rate, defect densities and bonding strength at the defects, have been attempted to simulate the alteration in the cluster morphology and density after annealing. Nevertheless, all the results show an Ostwald ripening despite the ripening starts at separate temperature. The kinetic processes alone do not yield a comparable alteration as observed in Figure 3; the results exhibit neither a decreased size nor an increased cluster density at elevated temperature. Figure 5 demonstrates the most similar size evolution with temperature whereas the decreased size of clusters at 450 K did not appear as in the experiments. The above kinetic model, a constrained Ostwald ripening, does not account exclusively for the present observation. We therefore argue that the atypical alteration of cluster size and density is associated with an energetics effect; an energetically preferential size might exist, although it could not be determined properly in the present study.

### 2.4. The Effect of Annealing-Induced Morphological Alteration on Catalytic Reactions

The thermally induced alteration in the morphology of Rh clusters on $Al_2O_3$/NiAl(100) can be applied in promoting catalytic reactions. Through appropriate annealing, we may reduce the cluster size to enhance the surface area and reactivity (the production per surface Rh site) of the clusters. Earlier studies have shown that the reactivity of supported Rh clusters toward methanol decomposition depends on the cluster size; when the cluster diameter becomes similar to or smaller than 1.5 nm, the reactivity becomes enhanced [31]. We therefore examined the present effect with methanol decomposition. Methanol-$d_4$ was used for the series of TPD experiments, because adsorbed methanol and methanol-$d_4$ showed similar desorption behavior, but the latter gave $D_2$ signals clearer than $H_2$ ones. As the decomposition of methanol-$d_4$ on either as-prepared or annealed Rh clusters proceeded only through dehydrogenation, we monitored the reactivity with TPD spectra of $D_2$.

Figure 6a compares the $D_2$ TPD spectra from 2.0-L methanol-$d_4$ ($CD_3OD$) adsorbed on 0.5-MLe Rh clusters/$Al_2O_3$/NiAl(100) as prepared and annealed to 450 K. Both the line shape and integrated intensities, corresponding to the production of $D_2$ molecules, are resembling. As the cluster size and density altered little for 0.5-MLe clusters (mean diameter about 1.4 nm) annealed to 450 K, shown in Figures 1 and 3, it is rational that the production from decomposed methanol-$d_4$ differed little for the clusters of both kinds. Figure 6b shows a contrast—the $D_2$ spectra from 1.6-MLe Rh clusters. Their line shapes are similar but the integrated intensity for annealed clusters exceeds that for as-prepared clusters by about 40%, indicating that 40% more $D_2$ were produced. The production was enhanced by two factors—first, an increased total Rh surface area and second, an enhanced reactivity due to the decreased size. The annealing to 450 K altered little the mean height (about 0.76 nm) but decreased the mean diameter from 2.3 nm to 1.5 nm (Figure 3a); the decreased size increased the surface/volume ratio and also the cluster density (Figure 3d), which increased the total Rh surface area so the production. Additionally, as indicated in earlier works, the clusters with diameter <1.5 nm have an enhanced reactivity, because the concentration of reactive Rh corner sites increases in smaller clusters [31]. A considerable fraction of these annealed clusters must have a diameter <1.5 nm (Figure 3a); they had an enhanced reactivity so increased the production of $D_2$.

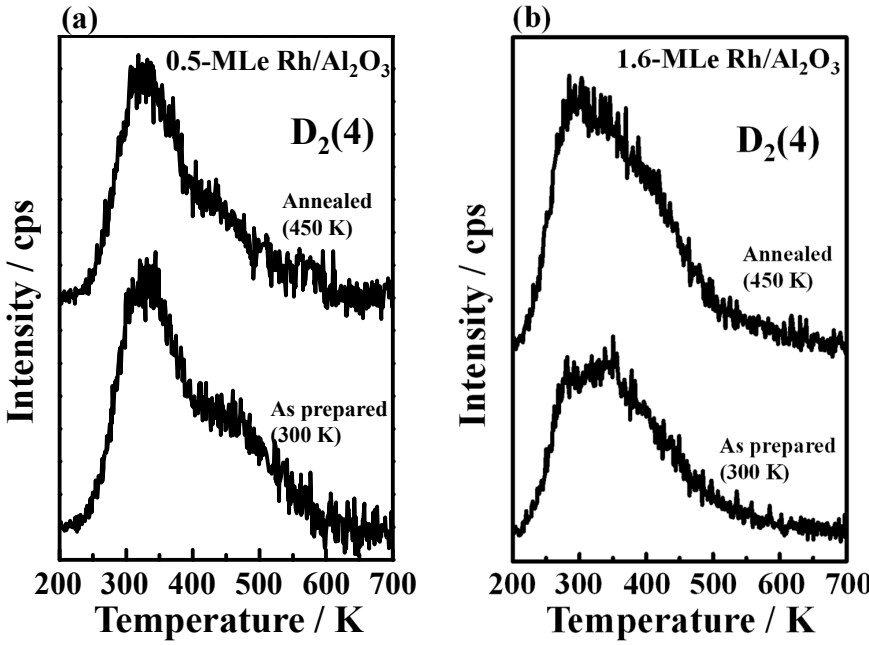

**Figure 6.** Temperature programmed desorption (TPD) spectra of $D_2$ (*m/z* = 4 u) from 2.0-L $CD_3OD$ adsorbed on (**a**) 0.5- and (**b**) 1.6-MLe Rh clusters/$Al_2O_3$/NiAl(100) as prepared at 300 K (lower panels) and annealed to 450 K (upper panels).

Our PES spectra of C 1s core level provide additional evidence for the promoted production. The spectral features for methanol on Rh clusters as prepared and annealed (450 K) are similar. Figure 7a exemplifies the C 1s spectra with those from 5.0-L methanol adsorbed on annealed (450 K) 1.0-MLe Rh clusters (mean diameter about 1.8 nm) on the alumina at 125 K and annealed to selected temperatures. As most multilayer methanol already desorbed at 125 K [31,55,56], monolayer methanol became predominant on the surface. The line initially about binding energy (BE) 287.4 eV is assigned to C 1s of methanol adsorbed on both the Rh clusters and alumina film; the feature about 285.0 eV is contributed by the C 1s signals of contaminative elemental carbon, according to earlier measurements for CO dissociation. With PES, we have measured the dissociation of CO into elementary carbon and oxygen on the Rh clusters. The dissociation rate depends on the cluster size and resembles earlier studies, showing that the rate varied between 0.25 and 0.5 [38]. Elevating the sample temperature to 175 or 225 K, adsorbed methanol either decomposed or desorbed; the remaining methanol, methoxy and CO from decomposed methanol ($CO_m$) on the Rh clusters gave a diminished C 1s feature about BE 287.4 eV. These species are indistinguishable in the PES spectra as their C 1s signals are at near BE [37,38,57]. Above 300 K, the C 1s line at 287.4 eV attenuated continuously and shifted positively to 287.6 eV. The BE shift is attributable to $CO_m$ which coexisted with atomic oxygen (O) or hydroxyl (OH)—the former came from dissociated $CO_m$ and the latter from O combining with H from dehydrogenated methanol [58–60]. The dissociation of $CO_m$ occurred actively above 300 K, indicated by enhanced C 1s signals for elemental carbon (285.0 eV) [37,38]. The C 1s signals of $CO_m$ disappeared near 500 K via dissociation or desorption of $CO_m$, which agrees with the above TPD spectra [31,56]. Although similar features were observed for Rh clusters as prepared, their evolution of C 1s intensities with temperature differed. Figure 7b compares the integrated intensities of C 1s of methanol/methoxy/$CO_m$ as a function of temperature from 1.0-MLe Rh clusters as prepared and annealed (450 K). The C 1s intensity at 125 K arose primarily from monolayer methanol but that above 250 K mostly from $CO_m$, according to earlier studies with infrared reflection absorption spectroscopy [31]; the decreasing rate with temperature of C 1s signals at 125–250 K (largely from absorbed methanol) is greater than that at 250–500 K (primarily from $CO_m$), because of a greater activation energy for CO desorption ($\geq$350 K) [31] or dissociation [37,38]. The ratio of the C 1s intensities at 300 and 125 K thus reflects that of $CO_m$ and

adsorbed methanol quantities. It is notable that the ratio ($CO_m$ vs. adsorbed methanol) increased on the annealed clusters: about 40% on the clusters as prepared whereas about 55% on the clusters annealed to 450 K. The increased ratio indicates an increased reaction probability of adsorbed methanol. The result corroborates that the clusters with diameter >1.5 nm have a decrease in size, as described above, after annealing to 450 K so an enhanced production of $CO_m$.

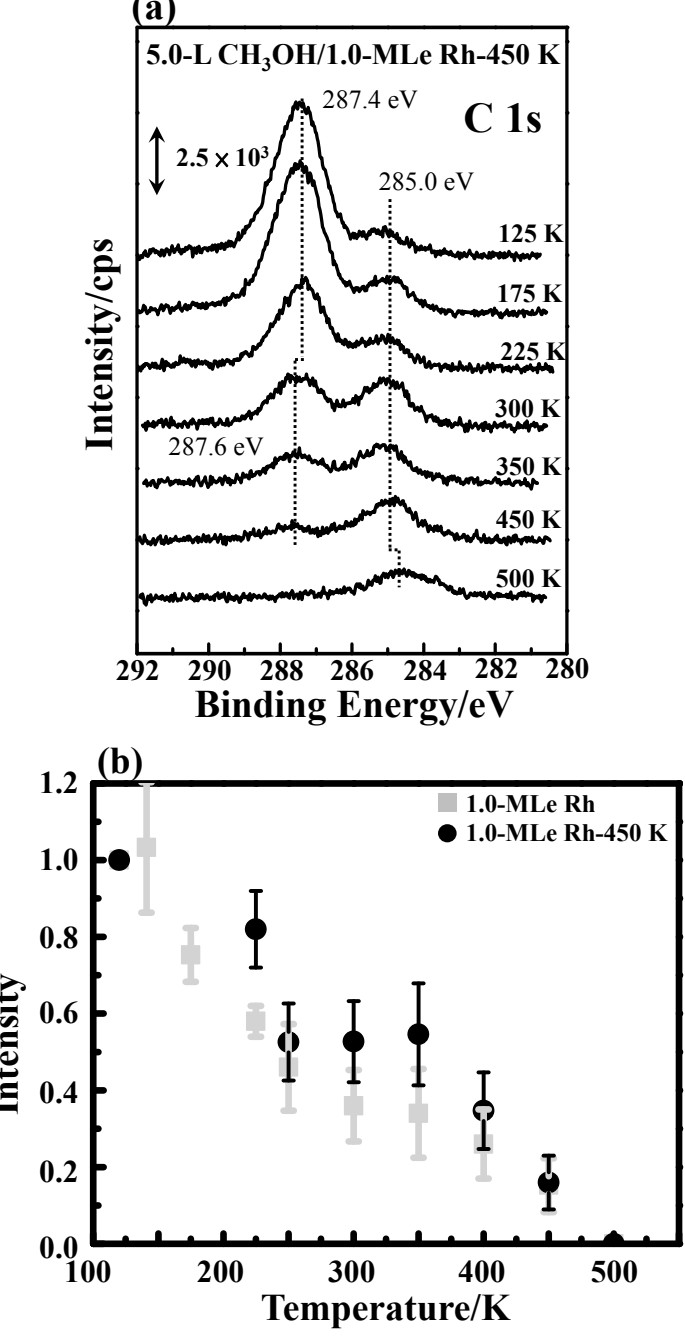

**Figure 7.** (**a**) C 1s photoelectron spectra from 5.0-L $CH_3OH$ adsorbed on 1.0-MLe Rh clusters on $Al_2O_3$/NiAl(100) at 125 K and annealed to selected temperatures; (**b**) integrated intensities of C 1s spectra for methanol/methoxy/CO from 1.0-MLe Rh clusters as prepared (300 K) and annealed (450 K) as a function of temperature. Each spectrum was recorded when the surface was cooled to about 125 K after annealing to the indicated temperature. For (**b**), the error bars indicate reproducibility and errors from fitting to derive the intensities. The intensity of the C 1s line for methanol/methoxy/CO obtained from any case was normalized to the intensity at 125 K.

This property implies a facile approach, though appropriate annealing, to prepare small Rh clusters; such small Rh clusters have a greater surface/volume ratio and could also be more reactive toward catalytic reactions. Moreover, the result implies also that such supported Rh clusters can tolerate an elevated reaction temperature to a certain extent: the Rh clusters can resist sintering about 450 K.

## 3. Materials and Methods

### 3.1. Experimental Details

We conducted the experiments in UHV chambers with a base pressure near $10^{-10}$ Torr. The NiAl(100) single crystal was purchased from MaTeck GmbH, Jülich, Germany; it had a roughness less than 30 nm and an orientation accuracy better than 0.1°. We performed alternative cycles of sputtering and subsequent annealing to clean the sample surface before each experiment. The cleanliness of the sample was examined with surface probe techniques such as STM, low-energy electron diffraction and Auger electron spectroscopy. The ultrathin alumina film was formed on oxidation of the NiAl(100) surface at 1000 K. To have a homogeneous crystalline oxide surface, we conducted post-oxidation annealing for the sample. The grown alumina film had thickness 0.5–1.0 nm [42,50]. The sample was then cooled down to 300 K for deposition of Rh vapor produced from a Rh rod heated in an evaporator (Omicron EFM 3). The deposition rate of Rh was fixed near 0.15 ML/min, estimated in accordance with the Rh coverage prepared at 300 K. The Rh coverage was calculated with the volumes of the Rh clusters measured with STM; 1.0 monolayer equivalent (MLe) amounts to a density of $1.39 \times 10^{15}$ Rh atoms/cm$^2$ (fcc Rh(100) surface atoms). After the growth of Rh clusters, the sample was quenched to 100 K (unless specified) for adsorption of methanol-$d_4$ (methanol). Methanol-$d_4$ (methanol) gas was dosed with a background pressure of $2–5 \times 10^{-9}$ Torr and by a doser pointing toward the sample. The methanol-$d_4$ and methanol (Merck, Darmstadt, Germany, 99.8%) were further purified by several freeze-pump-thaw cycles. Their exposures are reported in Langmuir (1 L = $10^{-6}$ Torrs).

The STM measurements were performed with an RHK UHV 300 unit (RHK Technology, Troy, MI, US). The images (constant-current topographies) were typically obtained at 100 K with a sample bias voltage of 2.2–2.8 V and a tunneling current of 0.7–1.2 nA. The STM tip was made of a tungsten wire etched electrochemically. RHEED was conducted by using an incident electron beam (40 keV) at a grazing angle (2–3°) relative to the surface. TPD spectra were taken with a quadruple mass spectrometer (Hiden) to monitor various masses and by ramping the sample at a rate of 3 K/s; the spectrometer was shielded and placed near the sample (about 2 mm). We performed our PES experiments with the BL 09A2 spectroscopy beamline at National Synchrotron Radiation Research Center in Taiwan [61]. The beam, with a photon energy at 383 eV, was incident normal to the surface, whereas photoelectrons were collected at an angle 58° off from the surface normal. The energy resolution in the measurements was about 0.1 eV. All PES spectra presented in the current work were normalized to their photon flux. The BE is referred to Al 2p core-level at 72.9 eV of NiAl bulk [62–64].

### 3.2. Modelling Method

Monte Carlo simulations that model the kinetic processes of surface Rh were performed on a three dimensional cubic lattice. Rh atoms at the bottom layer were assumed to have bonding with the substrate. The cubic lattice was initially empty; atoms were gradually added into the lattice sites by randomly choosing an empty one either at the bottom layer or on top of the existing islands formed by nucleating atoms. Each Monte Carlo step involved random hopping for one of the atoms in the lattice.

Each Rh atom in the lattice had a total binding energy from two contributions $n\epsilon_1 + \epsilon_2$. The first $n\epsilon_1$ was the binding energy with neighboring atoms, with $n$ the count of neighboring occupied sites. The neighboring sites for a particular site were defined as the eight surrounding sites at the same layer and night close-by sites at the above and also underlying layers (except the bottom layer). This stacking is not exactly like the observed fcc structure but is convenient for the modelling to proceed and yield

comparable evolution of island morphology and density. The second term $\epsilon_2$ was the binding energy with the substrate, applicable to Rh atoms at the bottom layer. To model the substrate with linear defects which had greater binding energy for adsorbed atoms, as seen in the experiments, for the $100 \times 100$ bottom layer of our simulations, the lattice sites of index $(10m, k)$ with $m = 0, 1 \cdots 9$ and $k = 0, 1 \cdots 99$, had a larger substrate binding energy $h\epsilon_2$ and the others just $\epsilon_2$. For simplicity, we set $\epsilon_1$ and $\epsilon_2$ at the same value $\epsilon$ for our simulation.

A Monte Carlo step was defined as every atom being considered to undergo random hopping into neighboring empty sites. For each possible hopping, the difference $\Delta E$ between the binding energy of the new and old sites was calculated. The probability of making the hop was governed by the Boltzmann factor $e^{-\Delta E/kT}$ if $\Delta E > 0$, and the probability was one (making the hop) if $\Delta E \leq 0$. The adsorption process was similar to the experiments. An atom was added for every $M$ Monte Carlo steps, at an initial fixed temperature. Once the desired number of atoms was added, indexed as the monolayer (ML), more Monte Carlo steps were performed to reach a stable configuration. The temperature was then raised to a higher one for next cycle of Monte Carlo steps to attain a new stable state. The temperature could then be raised again and the process repeated.

The DFT calculations were performed to derive the activation energies ($E_a$) for the Rh hopping processes. The calculated $E_a$ allowed us to confirm that the relative rates of the various hopping processes used in the Monte Carlo simulation are reasonable. Details can be found in the Supplementary Materials.

## 4. Conclusions

In this work, STM, RHEED, TPD, PES and Monte Carlo simulations were used to investigate the thermally induced evolution of morphology, structure and reactivity of Rh nanoclusters on ultrathin alumina (θ-$Al_2O_3$(100)) film formed on NiAl(100) single crystal. The Rh clusters were grown by deposition of Rh vapors onto the alumina film at 300 K and annealed to selected temperatures up to 800 K. The mean diameter of the Rh clusters as grown evolved from 1.4 to 2.3 nm and their height from 0.6 to 0.8 nm when the coverage increased; they grew in an fcc phase and had their facets (100) parallel to the alumina surface. Relative to bulk Rh, their lattice expanded up to 6%, with decreased size of the clusters. After the annealing, the cluster orientation remained and the lattice parameter decreased only a little ($< 2\%$). In contrast, the cluster morphology responded in a dramatic manner to the elevated temperature. The size of small clusters (diameter ≤1.5 nm) altered little with the annealing, whereas that of the large ones (diameter ≥ 2.0 nm) changed remarkably—the mean diameter decreased to about 1.5 nm on annealing to 450 K, despite the height being sustained, but increased to above 2.0 nm at temperatures ≥570 K. An evident Ostwald ripening occurred only for large clusters annealed to 800 K. The atypical decrease in cluster size at 450 K results largely from an energetic effect. The decreased size enhanced the surface/volume ratio and also the reactivity of the clusters toward methanol reactions, so increased the production of $D_2$ ($H_2$) and CO from decomposed methanol-$d_4$ (methanol).

**Supplementary Materials:** The following are available online at http://www.mdpi.com/2073-4344/9/11/971/s1, Figure S1: the computed activation barriers ($E_a$) for a Rh single atom which diffuses on θ-$Al_2O_3$(100) surface, dissociates from and wags along (diffusion along a cluster edge) from the clusters on θ-$Al_2O_3$(100) surface.

**Author Contributions:** Conceptualization, M.-F.L. and P.C.; methodology, M.-F.L., P.C. and J.-H.W.; formal analysis, Z.-H.L., P.-W.H., T.-C.H. and Z.-Y.C.; investigation, Z.-H.L., P.-W.H., T.-C.H., G.-J.L., Z.-Y.C., Y.-L.L. and L.-C.Y.; writing—original draft preparation, Z.-H.L., P.-W.H., M.-F.L., P.C. and J.-H.W.; writing—review and editing, M.-F.L., G.-J.L., J.-H.W. and Y.-J.H.; supervision, M.-F.L., J.-H.W. and Y.-J.H.; funding acquisition, M.-F.L.

**Funding:** This research was funded by Ministry of Science and Technology in Taiwan, grant number MOST-103-2112-M-008-014-MY2.

**Acknowledgments:** The CPU time at Taiwan's National Center for High-performance Computing is greatly appreciated.

**Conflicts of Interest:** The authors declare no conflict of interest.

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
