# Peer review of "Investigation of Thermal Stability and Reactivity of Rh Nanoclusters on an Ultrathin Alumina Film"

_catalysts, doi:10.3390/catal9110971_

Round 1

Reviewer 1 Report

Dear Authors and Editor,

the manuscript is suitable for publication in Catalyst after minor revision.

Line 55: the Authors say that Ru grows as fcc nano-crystal. It must be specified that this crystal structure is NOT the same of bulk Ru.

Lines 74-78: the Authors must state what they mean for monolayer equivalent.

Line 380: the Authors cannot compare the lattice parameter of fcc Ru with the one of bulk Ru. Natural form of Ru is hexagonal (hcp) with a=b=270 pm, c=428 (font: webelements.com, highly trustable).

Figures:

all figures are readily readable and understandable, I did not notice copy&paste among the figures.

Bibliography:

really exhaustive.

My kind regards.

Reviewer 2 Report

The referee has read the paper with much interest. Despite the fact that the  manuscript is well written and organized, the reported results better fit to a materials science journal rather Catalysts. From the reviewer's point of view, the manuscript should be severely revised in order to be re-evaluated. The reader of Catalysts would expect a detailed discussion of catalytic phenomena with respect to the decomposition reaction.  Kinetic behaviour of the catalysts is not sufficiently studied. The implication of the kinetic performance with physicochemical properties should be more deeply studied, while a mechanistic consideration is the key issue and should be carefully studied.

Reviewer 3 Report

Small Rh nanoparticles were deposited on ultrathin Al2O3 layer (formed on polished 100 NiAl under ultrapure conditions) by evaporation, and subjected to thermal annealing at different temperatures. The change in lateral size, height and density was investigated, with the curious result that at a specific loading the particles first decrease in size before increasing with rising temperature. Finally, the materials were applied in the catalytic decomposition of methanol.

I have severe problems following the authors’ main point: The presence of a “magic” size around 1.5 nm towards which the particles gravitate. As the authors state themselves, their observation contradicts the expectation (Ostwald ripening with decreasing particle density and increasing size of the larger particles on cost of the small ones). Also, it contradicts their modelling, such as outlined in figure 5 and the corresponding discussion. Also, throughout the work, samples are used inconsistently, so that characterization is not provided for all of them (showing 0.5 and 1.35 MLs as STM; summarizing data additionally for 1.6 ML; later in the catalysis section in figure 7 working with 1.0 MLs).

To make things worse, the experimental series apparently exhibiting the initial particle size decrease shows a complex pattern. First, the particles shrink with increasing temperature, which is followed by the expected size increase at higher temperatures.

When breaking down which data in the manuscript actually support the particle size decrease towards 1.5 nm, it appears that the whole discussion relies on a single STM image:

1) The 0.5 ML Rh series does not show the size focussing feature at all.

2) The 1.35 ML series shows the unexpected decline in size before the increase. This is only true if comparing the pristine sample (figure 2a) with the first annealing temperature (figure 2b). Unfortunately, when comparing the quality of figure 2a with the other STM images, it appears particularly fuzzy. Considering the below-average quality of this image and the errors to be expected for such an analysis (see the quite huge error bars in figure 3), the main result of the work might just be a measurement artifact.

3) The 1.6 ML series is mentioned in the figure 3 summary, and shows a similar pattern, but no STM images are provided.

Also, I disagree with the widespread use of the term “cluster” in the manuscript. Clusters are usually associated with small numbers of metal atoms and a rather precise structure (e.g. required for filling geometric or electronic shells). Assuming spherical geometry, Rh nanoparticles of 1.2 nm diameter consist of about 88 atoms, which clearly is in the range of larger clusters. At 2 nm diameter, the atom number increases to about 406. Particles of such or even larger sizes, which are very common in the materials shown in the manuscript, should be more accurately described as small nanoparticles, not as clusters. The probable reason for the frequent use of the term cluster is to explain the observation with the 1.35 ML sample. However, particles with several 100 atoms are far from a state in which a certain size has extraordinary stability when comparing it with particles possessing a few more / less atoms. Such a pronounced local energy minimum would however be required for efficiently focussing the particle size towards it.

In summary, I do not consider the presented data sufficient and convincing enough to provide unambiguous evidence for the controversial magic cluster size explanation. Thus, I cannot recommend publication.

Reviewer 4 Report

General:

The manuscript submitted by Z-H Liao et al. („A preferential size of Rh nanoclusters ….”) reports the coverage dependent thermal behavior of Rh deposited on epitaxial Al2O3 ultrathin film. It provides a multitechnique (STM, RHEED, TPD, PES, MC simulation, DFT) characterization of the system which is in general a valuable complex contribution among the similar studies on the oxide supported metal nanoparticle model systems last years. Moreover, the work presents also some evidences for the CH3OH reactivity dependence of the Rh/Al2O3 catalyst as a function of the thermal treatment and nanoparticle size. In spite of these positive concerns of the manuscript, the main message of the work (which is pressed also in the title) not sufficiently convincing, namely, that a preferential size (1.5 nm) of the Rh nanoparticles exists and exhibits a special reactivity in the decomposition of methanol. My impression is that without the too-stressing of this conclusion on the special particle size and by applying an appropriate modification of the presented results, the paper will be publishable in CATALYSTS.

Detalied points for revising:

The presented STM measurements are crucial for the conclusion of the „preferential size of Rh nanoclusters”. The results for three different coverages (0.5 ML, 1.35 ML and 1.6 ML) are summerized in Figure 3 (although STM images are presented only for the first two cases), nevertheless , the main message for the „preferential size” (and the surprising decrease of the larger particles at 450 K) can be seen in Figure 03 (a) for the cases of 1.35 ML and 1.60 ML, but it is presented as images only for 1.35 ML in the Figure 2 (a)-(b); unfortunately, if we look carefully at these images, we can obviously perceive that the image (a) was taken up by a less sharp tip than the image (b) in Figure 2. This latter statement are also evidenced by the histograms related to these images. Concluding my objection against the „preferential size” cames from the experience that because of the strongly disturbing tip-particle shape convolution at these particle sizes, it is impossible to draw such a clear conclusion which is in the scope of the paper. Otherwise, if the authors insist on this statement they have to present stronger experimental evidences. For example, all the images should be presented in higher magnification (at least a double) or the particles of „preferencial size” should be imaged in even larger magnification in a sepatarate image. It is also an important conclusion of the paper that the top facet of the particles oriented in (001). Although this statement cames from the diffraction technique, nevertheless, it should be also seen by STM from the rectangular shape (outline) of the nanoparticles. Especially when only an individual particle is imaged by a very pointed tip. This type of image would strongly strenghten the paper. Concerning the enhanced reactivity of the „preferential size particles”, I feel also some weakness. It is generally accepted that the low coordinated Rh sites are more reactive for the dissociation of methanol and the probability of the existance of low coordinated sites is higher after the deposition at 300 K than following thermal treatment at 450 K. Consequently, the differences in the TPD curves (D2 evolution) in figure 6 do not deliver sufficiently strong evidence for the conclusion of the changed activity. Moreove, according to my estimation, the integrated intensity for the TPD curves in Figure 6 (b) do not differ by 40 %, only max 20%. Furtherly, I should ask the authors to assure the reproducibility of the curves presented in Figure 6 (b). In connection to the simulation (modelling) results, the authors conclude that a constrained Ostwald ripening does not account the experimental results (atypical alteration of the cluster size). I agree this conclusion, but I would complete with the fact, that the authors do not have strong arguments for this process even by the experiments (see above).

Some critical remarks about the text itself;

The last paragraph of the „Introduction” sounds like sentences fitting in the conclusions. I suggest to transfer and combine this part with the „Conclusions” chapter. At the same time, the Introduction should contain much more details about the literature concerning the particle size dependent activity in general. An expert extention of the „Introduction” would increase the quality of the paper. In some cases the english of the sentences are rather bad. Let me mention just an example. Instead of the present first sentence of the Conclusion I would write the following: „In this work STM, RHEED, TPD, PES and MC-simulation techniques were used for the investigation of thermally induced evolution of morphology, structure and reactivity of Rh deposited on ultrathin Al2O3 film formed on NiAl(100) single crystal surface.” Accordingly to my objections above, the title should be changed (for example : „Investigation of thermal stability and reactivity of Rh deposited on ultrathin Al2O3 film”).

Round 2

Reviewer 2 Report

Most of the criticism raised during first review, has been satisfactory covered by the authors. Despite the fact that the revised version still contains limited kinetic data, i can suggest publication

Reviewer 3 Report

Upon assessment of the performed changes, I still do not see full evidence for a cluster based mechanism of particle size focusing: Considering the error bars and the uncertainty of the size measurements, the magic size is not significant or at least unclear. The nanoparticles are present in distributions at all stages, a considerable reduction of the polydispersity is not observed.

Regarding my verdict, I adopt the position of referee 4, who is having very similar problems with this work. The manuscript heavily relies on the "stable cluster claim", which is not strongly supported by data. Clusters with magic numbers usually only contain several tens up to slightly above hundred atoms, are stabilized by a ligand shell, and present as colloid. This system here is totally different. The formation of such quite fragile, metastable structures by a quite rough temperature annealing step without ligand stabilization but on top of a support seems unlikely, or at least needs a huge attention to detail in order to investigate the impact of all these differing systemic parameters. This work would suit the journal with a different focus, under omission of the "magic cluster size" claim, or with new, crystal clear evidence. At the moment, neither is given, so I still cannot recommend publication.

Reviewer 4 Report

The corrected manuscript is worth for publishing in the present form.